# Analysis of treatment cost and persistence among migraineurs: A two-year retrospective cohort study in Pakistan

Kamran Khan[1☯], Mudassar Iqbal Arain[2☯], Muhammad Arif Asghar[1☯]*, Ahad Abdul Rehman[3☯], Muhammad Ali Ghoto[2☯], Abdullah Dayo[4☯], Muhammad Suleman Imtiaz[1☯], Mohsin Hamied Rana[5☯], Muhammad Asif Asghar[6☯]

**1** Department of Pharmaceutics, Institute of Pharmaceutical Sciences, Jinnah Sindh Medical University, Karachi, Pakistan, **2** Department of Pharmacy Practice, Faculty of Pharmacy, University of Sindh, Jamshoro, Pakistan, **3** Department of Pharmacology, Institute of Pharmaceutical Sciences, Jinnah Sindh Medical University, Karachi, Pakistan, **4** Department of Pharmaceutics, Faculty of Pharmacy, University of Sindh, Jamshoro, Pakistan, **5** Department of Research and Development, Reckitt Benckiser, Karachi, Pakistan, **6** Food and Feed Safety Laboratory, Food and Marine Resources Research Centre, PCSIR Laboratories Complex, Shahrah-e-Salimuzzaman Siddiqui, Karachi, Sindh, Pakistan

☯ These authors contributed equally to this work.
* m.arifasgher@hotmail.com

**Data Availability Statement:** All relevant data are within the paper and its Supporting Information files.

## Abstract

### Objectives

The persistence pattern of anti-migraine drugs' use among migraineurs is very low in the United States and different European countries. However, the cost and persistence of anti-migraine drugs in Asian countries have not been well-studied. Hence, the present study aimed to evaluate the treatment cost and persistence among migraineurs in Pakistan.

### Methods

Data from prescriptions collected from migraineurs who visited the Outpatient Department (OPD) of different public and private sector tertiary-care hospitals of Karachi, Pakistan were used to conduct this retrospective cohort study from 2017 to 2019. The minimum follow up period for each migraineur was about 12 months for persistence analysis while dropped-out patients data were also included in survival analysis as right censored data. Pairwise comparisons from Cox regression/hazards ratio were used to assess the predictors of persistence with the reference category of non-binary variables i.e. hazard ratio = 1 for low frequency migraineurs and NSAIDs users. Persistence with anti-migraine drugs was estimated using the Kaplan-Meier curve along with the Log Rank test.

### Results

A total of 1597 patients were included in this study, 729 (45.6%) were male and 868 (54.3%) were female. Non-steroidal anti-inflammatory drugs (NSAIDs) were the most prescribed class of drug initially for all classes of migraineurs (26.1%). Of them, 57.3% of migraineurs discontinued their treatment, 28.5% continued while 14.8% were switched to other

**Funding:** Reckett Benckiser provided support for this study in the form of salary for MHR. The specific role of these author are articulated in the 'author contributions' section. The funder had no role in study design, data collection and analysis, decision to publish, or preparation of the manuscript.

**Competing interests:** The authors have read the journal's policy and the authors of this manuscript have the following competing interests: MHR is a paid employee of Reckett Benckiser. There are no patents, products in development or marketing products to declare. This does not alter our adherence to PLOS ONE policies on sharing data and materials.

treatment approaches. Persistence with initial treatment was more profound in female (58.8%) patients compared to males while the median age of continuers was 31 years. The total cost of migraine treatment in the entire study cohort was 297532.5 Pakistani Rupees ($1901.1). By estimating the hazard ratios (HR) using the Cox regression analysis, it can be observed that patients with high frequency (HR, 1.628; 95%CI, 1.221–2.179; *p<0.0001*) migraine, depression (HR, 1.268; 95%CI, 1.084–1.458; *p<0.0001*), increasing age (HR, 1.293; 95%CI, 1.092–1.458; *p<0.0001*), combination analgesics (HR, 1.817; 95%CI, 0.841–2.725; *p = 0.0004*) and prophylaxis drugs (HR, 1.314; 95%CI, 0.958–1.424; *p<0.0001*) users were at a higher risk of treatment discontinuation. However, patients with chronic migraine (HR, 0.881; 95%CI, 0.762–0.912; *p = 0.0002*), epileptic seizure (HR, 0.922; 95% CI, 0.654–1.206; *p = 0.0002*), other comorbidities (HR, 0.671; 95%CI, 0.352–1.011; *p = 0.0003*) and users of triptan(s) (HR, 0.701; 95%CI, 0.182–1.414; *p = 0.0005*) and triptan(s) with NSAIDs (HR, 0.758; 95%CI, 0.501–1.289; *p<0.0001*) had more chances to continue their initial therapy.

## Conclusion

Similar to western countries, the majority of migraineurs exhibited poor persistence to migraine treatments. Various factors of improved persistence were identified in this study.

## Introduction

Migraine is a chronic and disabling disorder of neurovascular etiology, characterized by recurrent episodic headache attacks and a variable presentation among subjects. Treatments with migraine-specific drugs such as ergotamines or triptans (serotonin agonists) are profoundly efficient. However, more than one third of patients are triptan non-responders due to the high cost of treatment and this is the main reason for low persistence with the therapy [1]. Hence, nonspecific drugs such as nonsteroidal anti-inflammatory drugs (NSAIDs), combination analgesics, opioid analgesics, and non-opioid analgesics are frequently used [2]. According to the estimation of the World Health Organization (WHO), 324 million people worldwide have suffered from various classes of migraine [3]. In Pakistan, since 1990, migraineurs have increased by 14.6% with an average of 0.6% a year [4].

Evidence suggested that oral drugs demand strict adherence to treatment to obtain appropriate outcomes of therapy [5]. The patient's non-compliance has been found to cause serious problems in treating various chronic disorders. Similarly, non-persistence with migraine treatment has been recognized as an important health issue that enhances the additional health care costs per year [6].

Previously, costs, adherence and persistence with migraine medications have been reported using various methods in different countries [7–13]. The data reflected low persistence in migraineurs i.e. 26% to 29% and 17% to 20% at six- and twelve-month studies, respectively in the United States [14]. Moreover, according to the study conducted in Asia 34.3% of patients persists to migraine specific therapy while 40.9% of migraineurs switched to other class of anti-migraine drugs [15]. Serious adverse effects, long term treatment, self-medication and high cost of migraine specific drugs are the major factors of patients' non-persistence with therapy, which can cause enhancement in severity and relapse of migraine attacks [16, 17].

In our previous study, we reported the prescribing patterns of anti-migraine drugs concerning general physicians (GPs) and neuro physicians (NPs) in Southern Pakistan [18]. However,

according to our literature study, in Pakistan, no such study was conducted regarding the patient's persistence and cost analysis of anti-migraine therapy. Therefore, the study aimed to assess the cost of anti-migraine drug therapy used in the last two years of treatment in our study patients and analysis of the persistence of migraineurs towards the same.

## Methods

### Study design

This cohort study was performed from October 2017 to September 2019. The way of prescriptions collection from migraineurs has been defined in our previous study [18]. Similarly, in this study migraine patients received different treatment protocols such as NSAIDs alone, analgesics in combination, triptans, triptans with NSAIDs and some prophylaxis drugs like flumazenil, topiramate, propranolol, valproic acid, amitriptyline and gabapentin. Migraineurs are classified into three categories basis migraine attack frequencies, i.e. "Low" with less than 10 days of headache per month, "High" with 10 to 14 days of headache per month and "Chronic" with more than 15 days of headache per month [19].

### Data source

The ethical review board of Civil Hospital located in Karachi, Pakistan, approved data collection efforts for this study. The prescriptions were collected from migraineurs who visited the Outpatient Department (OPD) of different four public and four private sector tertiary-care hospitals of Karachi, Pakistan. The list of hospitals used for data collection in this study is given in **S1 Table**. In these hospitals, migraineurs belonging to every class of socioeconomic status frequently visited both the GPs and NPs. For investigation purposes, the sample size for this study was calculated using statistical software Open Epi (Version 2.3.1), keeping the anticipated frequency of 85% at a 95% confidence interval (CI) with a 5% margin of errors [18]. The collection of prescriptions from migraineurs was performed with their consent using a convenient sampling technique. All cases of adult migraineur patients, irrespective of gender and ethnicity, who visited the hospitals, were selected for this study on the following specific criteria including (a) patients who gave permission & were willing to join this study and (b) a confirmed diagnosis of migraine and at least one episode of migraine during last one month. However, patients who were completed the minimum followed up period of 12 months or had an experienced of any event (discontinuation or switches) during this period were included for the determination of treatment persistence i.e. 273 days continuation. As exclusion criteria, people who used homeopathic and herbal medications, performed cupping therapy and pregnant women were not included. Demographic and disease data of patients were collected from the previous case profiles of patients without their interviews. Besides, we also collected duplicate prescriptions from a bearer of prescriptions who visited the hospital. The data written on each prescription included hospital name, patient's name, age and gender, diagnosis, medication regimen, duration of therapy and follow up for the next visit.

### Analysis of persistence

Persistence with treatment was defined as the duration of treatment of more than 273 days from the initial drug prescription [20]. All patients were divided into three classes, continuers, switchers and discontinuers. Patients who continued with the prescribed medication by practitioners in our specific study period (allowable 30-days gap) were considered as continuers. Patients who changed to another class of drug from initially prescribed drug(s) were called

switchers while patients who dropped out from the therapy were categorized as discontinuers [21].

## Cost analysis

The direct cost of anti-migraine drugs from prescriptions was calculated and expressed as the annual cost of drugs for continuers, switchers and discontinuers individually. The costs of each prescription are presented into Pakistani Rupees (PKR) and United States Dollars (USD) i.e. $1 = 156.50 Rupees (Rs.).

## Statistical analysis

All extracted data from patient prescriptions are presented as their median ± interquartile range (IQR). Analysis of statistical significance was performed on continuous and non-continuous variables using one-way anova and Chi-square test respectively on SPSS (version 23) software. Multiple risk factors with respect to the patient's persistence to therapy were analyzed using Cox regression analysis using the NCSS statistical software (version 20). The Cox proportional hazard regression model is often used to analyze covariate information that changes over time, with the hazard proportional. Therefore we used the data of all study patients including the data of patients who drop-out during study due to any reason as right censored data in survival analysis. In addition, without taking into account any possible confounding covariates, the cumulative proportion of patients persisting with their initial prescription drug of each study cohort was estimated using the Kaplan–Meier method along with Log Rank test. All p-values are two-tailed with $p < 0.05$ and $p < 0.0001$ level of significance.

## Results

A total of 2043 migraineurs were enrolled for this study. Out of these, 354 (17.3%) participants were excluded due to multiple reasons as given in **Table 1** while 92 patients' data were censored due to an incomplete follow-up period. Thus, 1597 migraineurs were included in this study, 729 (45.6%) males and 868 (54.3%) females. The mean follow up period for each migraineur was 457 ± 72.3 days with the range of 229 to 686 days.

**Table 1. Summary of dropout patients with reasons at different stages of study.**

| Study stage | Reasons of drop-out | N (%) |
|---|---|---|
| Initial screening or Pre-assessment | Used homeopathic or herbal medications | 95 (4.65) |
| | Performed cupping therapy | 57 (2.79) |
| | Pregnant women | 20 (0.97) |
| | Used multiple anti-migraine drugs | 59 (2.88) |
| | Used other class of anti-migraine drugs | 101 (4.94) |
| | Not provide complete data of drug regimen | 22 (1.07) |
| During or end of study | Died or moved away | 41 (2.00) |
| | Pregnancy | 5 (0.24) |
| | Follow-up period of less than 12 months and not experienced any event during this period | 46 (2.25) |

Total participants enrolled initially = 2043 N

Total drop out patients = 446 N (21.8%)

Data of patients who drop-out during the study period due to any above defined reasons were censored in survival analysis

**Table 2. Demographics and characteristics data of migraineurs by class of anti-migraine drugs.**

|  | NSAIDs | Combination analgesics | Triptans | Triptans + NSAIDs | Prophylaxis drugs | P-value |
|---|---|---|---|---|---|---|
| **Migraineurs** [N (%)] | | | | | | |
| Low Frequency | 263 (16.5) | 235 (14.8) | 38 (2.3) | 22 (1.3) | 92 (5.8) | 0.015 |
| High frequency | 109 (6.8) | 77 (4.8) | 106 (6.6) | 127 (8.0) | 60 (3.7) | |
| Chronic | 43 (2.7) | 68 (4.2) | 127 (8.0) | 194 (12.2) | 36 (2.2) | |
| **Total** | 415 (26.1) | 380 (23.9) | 271 (17.0) | 343 (21.6) | 188 (11.8) | |
| **Median age** [yrs (±IQR**)] | 26 (8–34) | 27 (8–38) | 29 (11–43) | 30 (10–54) | 33 (12–47) | 0.035 |
| **Gender** (%) M/F | 46.1/53.9 | 53.7/46.3 | 38.7/61.3 | 34.4/64.6 | 53.3/46.5 | 0.038 |
| **Patients with depression** [N (%)] | 40 (2.5) | 49 (3.0) | 29 (1.8) | 42 (2.6) | 129 (8.1) | 0.007 |
| **Epilepticus seizures** [N (%)] | 28 (1.7) | 41 (2.5) | 22 (1.3) | 19 (1.1) | 234 (14.7) | 0.002 |
| **Patients with other comorbidities** [N (%)] | 54 (3.4) | 24 (1.5) | 39 (2.4) | 20 (1.2) | 152 (9.5) | 0.032 |

*Total (1597 N) Patients

**Interquartile range

The median age of participants was 35 years (IQR 9–54 years). NSAIDs were the most prescribed class of drug (26.1%), followed by analgesics in combinations (23.9%), triptans with NSAIDs (21.6%), triptans alone (17.0%) and prophylaxis drugs (11.8%) (**Table 2**). Demographics and characteristic data of migraineurs by a class of anti-migraine drugs are also presented in **Table 2.**

Within the study population, it was observed that discontinuers ranked first with high percentages (57.3%) followed by continuers (28.5%) and switchers (14.8%) (**Table 3**). Among all, chronic migraineurs shared the highest percentage of continuers (18.8%) whereas the least percentage of low frequency migraineurs (2.9%) continued their initial regimen throughout the study follow-up period. Demographic and characteristic data with respect to persistence patterns of migraineurs are also given in **Table 3.** Female participants were found to be having more persistence *(p = 0.014)* with initial treatment compared to males.

Patterns of persistence with different classes of anti-migraine drugs are presented in **Table 4.** Migraineurs who were using triptans alone (57.5%) or triptans with NSAIDs (54.5%) initially showed much more tendency to stay on anti-migraine treatment (continuers) compared to those who used other classes of drugs initially. The Kaplan–Meier curves for therapy

**Table 3. Demographics and characteristics data by persistence patterns of migraineurs.**

|  | Continuers | Switchers | Discontinuers | P-value |
|---|---|---|---|---|
| **Migraineurs** [N (%)] | | | | |
| Low Frequency | 46 (2.9) | 84 (5.2) | 520 (32.8) | 0.002 |
| High frequency | 107 (6.7) | 94 (5.9) | 278 (17.5) | |
| Chronic | 299 (18.8) | 58 (3.6) | 111 (7.0) | |
| **Total** | 452 (28.5) | 236 (14.8) | 909 (57.3) | |
| **Median Age** [yrs (±IQR**)] | 31 (9–50) | 27 (10–42) | 36 (13–54) | 0.021 |
| **Gender** (%) M/F | 41.2/58.8 | 59.2/40.8 | 54.3/45.7 | 0.014 |
| **Patients with depression** [N (%)] | 19 (1.1) | 73 (4.5) | 60 (3.7) | 0.041 |
| **Epilepticus seizures** [N (%)] | 60 (3.7) | 17 (1.0) | 23 (1.4) | 0.028 |
| **Patients with other comorbidities** [N (%)] | 61 (3.8) | 65 (4.1) | 19 (1.1) | 0.034 |

* Total (1597 N) Patients

**Interquartile range

**Table 4. Persistence patterns of different class of anti-migraine drugs.**

| | Continuers | Switchers | Discontinuers | P-value |
|---|---|---|---|---|
| | N (%) | N (%) | N (%) | |
| **NSAIDs** | 36 (8.6) | 62 (14.9) | 317 (76.3) | 0.001 |
| **Combination analgesics** | 41 (10.7) | 70 (18.4) | 269 (70.7) | <0.001 |
| **Triptans** | 156 (57.5) | 38 (14.0) | 77 (28.4) | 0.025 |
| **Triptans + NSAIDs** | 187 (54.5) | 50 (14.5) | 106 (30.9) | 0.038 |
| **Prophylaxis drugs** | 32 (17.0) | 16 (8.5) | 140 (74.4) | 0.002 |
| **Total** | 452 (28.5) | 236 (14.8) | 909 (57.3) | <0.001 |

* Total (1597 N) Patients

continuation also showed that patients receiving triptans or triptans with NSAIDs had a significantly higher adjusted cumulative probability of remaining on the initial anti-migraine treatment compared with other drugs (**Fig 1**). During the first 3 months after the index date, 100% persistent behavior was observed in triptans users while treatment persistence was continuously lower in the NSAIDs, combination analgesics and prophylaxis drugs users. However, NSAIDs users showed a greater discontinuation percentage (76.3%) in comparison with other drugs. Switchers (18.4%) were more substantial in migraineurs who were given combination analgesics. In addition, the result of the Log Rank test also indicated that there were significant differences *(P<0.003)* were observed in treatment persistence among users of different anti-migraine therapy.

The median cost of drugs belonging to 5 different categories of drugs, weighted for the dose of drug recommended per day for each patient, was Rs.11.2 for NSAIDs, Rs.20.7 for combination analgesics, Rs.157.6 for triptans, Rs.168.0 for triptans with NSAIDs and Rs.18.1 for prophylaxis drugs. The total cost of migraine treatment in the entire study cohort was Rs. 297532.5 ($1901.1), which for continuers was Rs.153951.0 ($983.7); for switchers was

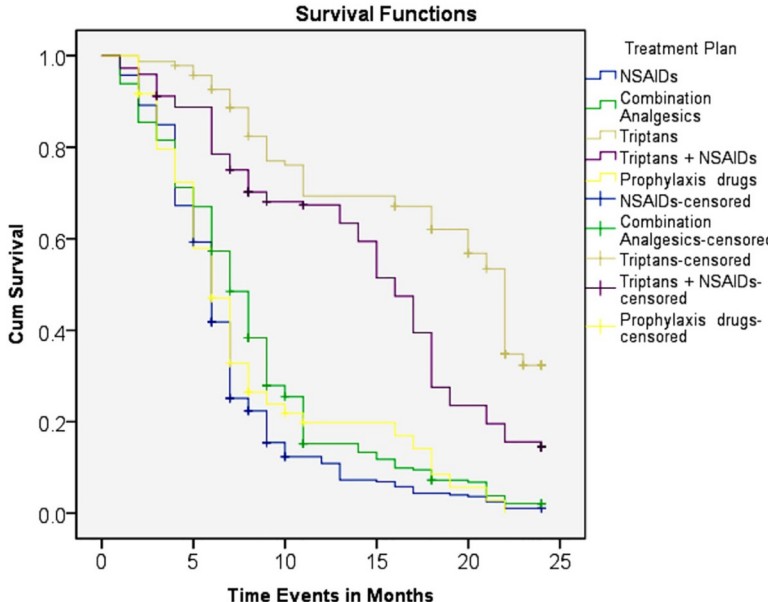

**Fig 1. Kaplan–Meier curves showing an adjusted cumulative probability of treatment persistence for 24 months after by anti-migraine drugs.**

**Table 5. Annual treatment cost of each migraine patient with respect to a different class of anti-migraine drugs.**

| Migraine therapy | Median cost for Continuers | | Median cost for Switchers | | Median cost for Discontinuers | | Median cost for study cohort | | P-value |
|---|---|---|---|---|---|---|---|---|---|
| | PKR[1] ± IQR[2] | USD[3] ± IQR | PKR ± IQR | USD ± IQR | PKR ± IQR | USD ± IQR | PKR ± IQR | USD ± IQR | |
| NSAIDs | 4837 ± (324–6459) | 30.9 ± (2.0–41.2) | 9232 ± (782–13641) | 58.9 ± (4.9–87.1) | 1181 ± (110–1478) | 7.5 ± (0.7–9.4) | 6824 ± (783–9457) | 43.6 ± (5.0–60.4) | <0.001 |
| Combination analgesics | 9046 ± (1420–12765) | 57.8 ± (9.0–81.5) | 26520 ± (9342–32788) | 169.4 ± (59.6–209.5) | 3055 ± (722–4267) | 19.5 ± (4.6–27.2) | 25756 ± (8644–34611) | 164.5 ± (55.2–221.1) | 0.004 |
| Triptans | 56823 ± (8129–74573) | 363.0 ± (51.9–476.5) | 7348 ± (2052–11753) | 46.9 ± (13.1–75.0) | 9211 ± (1890–13653) | 58.8 ± (12.0–87.2) | 36091 ± (9566–51467) | 230.6 ± (61.1–328.8) | 0.006 |
| Triptans + NSAIDs | 58774 ± (9266–74785) | 375.5 ± (59.2–477.8) | 32558 ± (6705–45345) | 208.0 ± (42.8–289.7) | 13676 ± (1876–17543) | 87.3 ± (11.9–112.0) | 47754 ± (8671–62785) | 305.1 ± (55.4–401.1) | 0.002 |
| Prophylaxis drugs | 7934 ± (780–12793) | 50.6 ± (4.9–81.7) | 11466 ± (1009–15786) | 73.2 ± (6.4–100.8) | 6389 ± (590–12764) | 40.8 ± (3.7–81.5) | 10056 ± (844–15775) | 64.2 ± (5.3–100.7) | 0.004 |
| Total Median Cost | 11267 ± (3732–18790) | 71.9 ± (23.8–120.0) | 12445 ± (8922–18897) | 79.5 ± (57.0–120.7) | 5263 ± (732–8642) | 33.6 ± (4.6–55.2) | 22674 ± (7764–31567) | 144.8 ± (49.6–201.7) | <0.001 |
| **Total cost** | **148931.2** | **951.6** | **105823.1** | **676.1** | **34992.3** | **223.5** | **145127** | **927.3** | **<0.001** |

[1]Pakistani rupees

[2]Interquartile range

[3]United States Dollar

[4]USD = 156.50 PKR (At: 25-Sep-2019)

Rs.108429.0 ($692.8) and for discontinuers was Rs.35152.2 ($224.6) with associated percentage shares of 51.7%, 36.4% and 11.8%, respectively (**Table 5**). The annual median cost of migraine treatment was Rs.22674 ($144.8). Based on initially prescribed class of drugs, the annual median costs were Rs.6824 ($43.6) for NSAIDs, Rs.25756 ($164.5) for combination analgesics, Rs.36091 ($230.6) for triptans, Rs.47754 ($305.1) for triptans with NSAIDs and Rs.10056 ($64.2) for prophylaxis migraine drugs. With respect to persistence patterns, the annual median cost of continuers was Rs.11267 ($71.9), Rs.12445 ($79.5) for switchers and Rs.5263 ($33.6) for discontinuers.

The statistical chi-square analysis showed that statistically significant relationships were found in the patient's treatment persistence with respect to all study variables and classes of anti-migraine drugs at $P<0.05$. By applying the Cox regression analysis, it can be observed that patients with high frequency (HR, 1.628; 95%CI, 1.221–2.179; $p<0.0001$) migraine, depression (HR, 1.268; 95%CI, 1.084–1.458; $p<0.0001$) and increasing age (HR, 1.293; 95%CI, 1.092–1.458; $p<0.0001$) were at higher risk of treatment discontinuation (**Table 6**). Alike, users of analgesics in combination and prophylaxis drug users were also at higher risk of discontinuation with the hazard ratios of 1.817 and 1.314, respectively. However, chronic migraineurs (HR, 0.881; 95%CI, 0.762–0.912; $p = 0.0002$), epilepticus seizure (HR, 0.922; 95%CI, 0.654–1.206; $p = 0.0002$), other comorbidities (HR, 0.671; 95%CI, 0.352–1.011; $p = 0.0003$) and triptans alone (HR, 0.701; 95%CI, 0.182–1.414; $p = 0.0005$) or in combinations with NSAID (HR, 0.758; 95%CI, 0.501–1.289; $p<0.0001$) users had more chances to continue their initial therapy.

## Discussion

In the present study, we analyzed the costs of antimigraine drug treatment(s) and how long the migraineurs persist on various antimigraine therapies. The use of NSAIDs and

**Table 6. Statistical analysis of migraineurs persistence towards treatment using chi-square and Cox proportional hazard regression analysis.**

| Risk factors | Cox Regression Analysis | | |
|---|---|---|---|
| | *P-value* | HR[3] | CI[4] (95%) |
| **Migraineurs** | | | |
| Low Frequency | 0.0002 | –– | |
| High frequency | <0.0001 | 1.628 | (1.221–2.179) |
| Chronic | 0.0002 | 0.881 | (0.762–0.912) |
| **Age** | <0.0001 | 1.293 | (1.092–1.458) |
| **Patients with depression** | <0.0001 | 1.268 | (1.084–1.458) |
| **Epilepticus seizures** | 0.0002 | 0.922 | (0.654–1.206) |
| **Patients with other comorbidities** | 0.0003 | 0.671 | (0.352–1.011) |
| **Class of drugs** | | | |
| NSAIDs | <0.0001 | –– | |
| Combination analgesics | 0.0004 | 1.817 | (0.841–2.725) |
| Triptans | 0.0005 | 0.701 | (0.182–1.414) |
| Triptans + NSAIDs | <0.0001 | 0.758 | (0.501–1.289) |
| Prophylaxis drugs | <0.0001 | 1.314 | (0.958–1.424) |

[1]Continuers, [2]Discontinuer

[3]Hazard Ratio

[4]Confidence interval, p<0.0001

*Figures in parentheses indicate the reference category of categorical variables (hazard ratio = 1), hazard ratio is adjusted for other variables in the table.

*Data of patients who drop-out during the study period or didn't experienced event during whole study period were censored in survival analysis.

combination analgesics was found to be more preferred by low and high frequency migraineurs than chronic migraine patients. This may be due to the fact that NSAIDs are more effective to relieve pain in mild to moderate conditions while less efficacious in chronic migraine compared to triptans [22]. Analysis of demographic data of patients showed that older subjects were more likely to use triptans alone or triptans with NSAIDs whereas a high usage ratio of NSAIDs was associated with younger migraineurs. The reasons may derive from multiple factors: chronic migraine allied with older age, short term or required early relief for younger migraineurs, and others [23]. It has been reported that female subjects were more persisted with NSAIDs than males which were also observed in this study [24]. Migraine is frequently associated as a comorbid with other disorders and epidemiological studies reported the high prevalence of epilepsy, stroke and psychiatric problems including depression, anxiety and mania in migraineurs [25]. Patients with depression and epilepsy were prescribed higher percentages of other classes of drugs i.e. divalproex sodium and topiramate, which showed better management of migraine with such comorbid conditions [24].

Overall, more than 50% of the study population discontinued their medication–potential reasons could be drug related adverse events, poor efficacy, dubious migraine diagnosis, or diminished pain frequency. Moreover, it is reported that half of the migraine patients discontinued their treatment without consultation with healthcare professionals [1]. Patients with chronic migraine showed more persistence and treatment continuity with anti-migraine drugs. Triptans and their combinations with NSAIDs are preferred treatments for chronic migraine which might be due to fewer side effects of triptans and better tolerability with increasing age of patients [25]. However, few continuers showed inconsistency in their

persistence behavior with migraine therapy specifically observed in triptans users and they restarted their therapy multiple times after pain relapse, which might be due to the initial pain relief gained and also the relatively high cost of triptans. Simply stated, patients who respond quickly to treatment and who do not experience recurrence require no additional care or drugs for time being. Those patients who achieved migraine relief with initial one to two week persistence with treatment, which would be expected to be the most effective and lowest cost approach. Nevertheless, if there is also a high recurrence after relapse, then total costs to treat the attacks will increase because it is expected that more than one dose will be taken for those who were initially successful but who recurred needs more proportion of dose. Self-medication may be a contributing factor associated with premature discontinuation of treatment in low frequency migraine compared with high frequency and chronic migraine. Given the consequences of early discontinuation and non-persistence, it is very important to enhance the treatment persistence to drug therapy which could reduce the relapse of disease and ultimately cost of the treatment. The patient's age showed an influence on drug persistence in this study. Subjects with a median age of 30 years were found to be more persistent with their treatment while the median age for those who discontinued their treatment was 36 years. Liu et al in 2011 and Etemad et al in 2005 reported similar observations that an increase in patients' age leads to treatment discontinuation [26, 27]. This might be since migraine prevalence is found to be reduced in older subjects [28]. This study demonstrated that females were more persistent with their treatment in comparison with males and these findings are in line with previous studies [8]. Male subjects in Asian cultures might not continue their treatment until their conditions have severely deteriorated; therefore, male subjects might have a low persistence ratio with anti-migraine therapy [29]. Patients with different comorbidities showed a higher ratio of switching to other drugs because there were more treatment complications associated with migraine comorbidities [30]. Adjustment of a single drug for two comorbid conditions is often difficult i.e. dose of a drug required for the treatment of migraine may be insufficient to treat the associated comorbid situation [31].

In our study, the discontinuity with all classes of anti-migraine was observed i.e. 57.3% which is almost similar in comparison with previously reported data that claims 54.9% of migraineurs completely discontinued all anti-migraine drugs [32]. Persistence with triptans was found to be appreciable (57.5%) and almost double the reported data in UK, France and Germany i.e. 14.6%, 14.7% and 13.7%, respectively [33]. This could be due to multiple reasons; firstly, treatment by specialists (neurologists) gives confidence to migraineurs to rely on the initial treatment plan especially with triptans. Secondly, confidence on neurologists and complete satisfaction with the efficacy of triptans to eliminate symptoms related to migraine.

In this study, we reported the data of both prevalent and incident patients. The main reason behind to evaluate the treatment persistence behavior for prevalent and incident patients simultaneously is to broaden the outcomes of this study. Moreover, in acute therapy like in acute migraine, it is necessary to evaluate the usage of prophylaxis drugs. However, persistence with prevalent patients showed different results in comparison with incident patients and approximate 70% of prevalent patients exhibit non-persistence behavior with their treatment which could be due to avoid some potential hazards associated with drug and one study concluded the same results of poor persistence with prevalent / prophylactic treatment [9].

The switching pattern and discontinuity of anti-migraine drugs have also been associated with the economic burden. Significant differences were observed in the total cost of treatment among continuers, switchers and discontinuers with respect to each class of anti-migraine drugs. These differences in cost were found due to the high ratio of switching of drugs from one class to another. Treatment persistence should be improved by choosing the more appropriate drug resources to avoid the long term complications and conversion of patients in a

chronic state [34]. Assessing the mean cost of migraine therapy is a key feature to evaluate the cost-effectiveness of alternative pharmacologic agents in migraine treatment. The overall cost of anti-migraine drug treatment was greatly affected by the class of drugs, age and persistence pattern. The mean cost of migraine treatment was high in the case of triptans and its combination with analgesics in comparison with other classes of drugs. Medications used in migraine treatment are expensive, especially triptans being the most expensive class of anti-migraine pharmacotherapy [27]. However, low cost does not mean that the drug has to be the first choice in all subjects because a patient's preference also depends on drug efficacy, safety and percentage tolerability in patients. Hence, chronic migraineurs showed greater continuity with triptans due to their efficacy and high tolerability. On the other hand, it has been observed that the cost of treatment was immensely reduced in those migraineurs who switched from triptans to other classes of anti-migraine drugs. It has been reported that such non-persistence with triptans and switching to other classes of drugs (NSAIDs and combination analgesic) has been massively associated with increased cost in primary care [1]. Consistent with previous studies, we also found that the overall cost of continuers was much higher compared to switchers and discontinuers [27, 35].

Statistical analysis using Chi-square test at 0.05 significance ratio showed that all studied factors produced a significant impact on patient's persistence with anti-migraine treatment. Hazard ratios (HRs) with 95% CI using Cox regression or hazard ratio analysis was used to evaluate the associations of studied variables with treatment persistence and time to treatment discontinuations. The larger the HR was, the higher were the chances of discontinuation of antimigraine drugs. The analysis showed that subjects with chronic migraine, epilepticus seizures have low HR in comparison with low and high frequency migraineurs which indicates greater persistency with treatment. However, chances of discontinuation were observed with increasing age due to low migraine prevalence in older patients. Confidence in the ability of triptans to reduce migraine pain and improve quality of life plays a major role in persistence with triptans.

## Limitations

Final considerations are much essential part of this discussion concerning the limits of this study. There are various anti-migraine drugs administered orally for migraine prophylaxis. In this study only five common classes of oral agents were reviewed. However, these five classes of anti-migraine drugs are the commonly utilized drugs and thus may be somewhat representative of the whole anti-migraine class of drugs. Polling of results across the study design may not fully be appropriate, which is also an important point of this study such as geographic and regional population, dosage of drugs, etc. Due to this limitation, we refrained from combining the results of observational studies. Moreover, fluctuation in the severity of migraine was found in many subjects or patients which may lead to termination of prophylaxis medication. On the point of fact, the study data is not fully warranted for patient persistence of discontinuation pattern. Also, this study did not use a new-user design, so it is unknown that how many patients previously exposed to migraine therapies before entering this study, which could be an important factor that impacts the persistence of patients. Despite all these limitations, we believe that our study findings providing a broad representation of treatment persistence with oral anti-migraine drugs and highlights the need of further studies to improve patient's adherence to migraine therapy.

## Conclusion

Observational studies support that oral anti-migraine drugs have a poor persistence ratio. New options with cost effective, improved tolerability and drugs with fewer dosing intervals may

improve patient's persistence to migraine therapy. Persistence with treatment should be considered as an endpoint in future observational studies exploring more concise pattern and usage of such therapies. In all chronic disorders, treatment persistence remains a big challenge and such conditions may require the use of drugs for a whole or major part of life.

## Supporting information

**S1 Table. List of hospitals used for data collection.**
(DOCX)

**S1 Data. Minimal data set (raw data).**
(ZIP)

**S2 Data. Supporting files for data calculation.**
(ZIP)

## Acknowledgments

The authors gratefully thank to the administration of different government and private hospitals of Karachi Pakistan for providing data access.

## Author Contributions

**Conceptualization:** Mudassar Iqbal Arain, Muhammad Arif Asghar.

**Data curation:** Kamran Khan, Muhammad Suleman Imtiaz.

**Formal analysis:** Kamran Khan, Muhammad Arif Asghar, Ahad Abdul Rehman, Muhammad Ali Ghoto, Abdullah Dayo, Muhammad Suleman Imtiaz, Mohsin Hamied Rana, Muhammad Asif Asghar.

**Investigation:** Kamran Khan, Ahad Abdul Rehman.

**Methodology:** Ahad Abdul Rehman.

**Project administration:** Mudassar Iqbal Arain, Muhammad Ali Ghoto, Abdullah Dayo.

**Supervision:** Mudassar Iqbal Arain, Muhammad Ali Ghoto, Abdullah Dayo.

**Validation:** Kamran Khan, Muhammad Ali Ghoto, Mohsin Hamied Rana, Muhammad Asif Asghar.

**Writing – original draft:** Kamran Khan, Muhammad Arif Asghar, Ahad Abdul Rehman, Muhammad Suleman Imtiaz.

**Writing – review & editing:** Muhammad Arif Asghar, Mohsin Hamied Rana, Muhammad Asif Asghar.

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
