## [Decision Letter · Decision Letter 0]

13 Jan 2021

PONE-D-20-33485

Analysis of treatment cost and persistence among migraineurs: A two-year retrospective cohort study in Pakistan

PLOS ONE

Dear Dr. Asghar,

Thank you for submitting your manuscript to PLOS ONE. After careful consideration, we feel that it has merit but does not fully meet PLOS ONE’s publication criteria as it currently stands. Therefore, we invite you to submit a revised version of the manuscript that addresses the points raised during the review process.

In particular, please strive to answer all queries completely, because this has to be the last round of reviews.

We look forward to receiving your revised manuscript.

Kind regards,

Claudia Sommer

Academic Editor

PLOS ONE

Journal Requirements:

We note that one or more of the authors are employed by a commercial company: Reckitt Benckiser.

3.1. Please provide an amended Funding Statement declaring this commercial affiliation, as well as a statement regarding the Role of Funders in your study. If the funding organization did not play a role in the study design, data collection and analysis, decision to publish, or preparation of the manuscript and only provided financial support in the form of authors' salaries and/or research materials, please review your statements relating to the author contributions, and ensure you have specifically and accurately indicated the role(s) that these authors had in your study. You can update author roles in the Author Contributions section of the online submission form.

3.2. Please also provide an updated Competing Interests Statement declaring this commercial affiliation along with any other relevant declarations relating to employment, consultancy, patents, products in development, or marketed products, etc.  

Reviewers' comments:

Reviewer's Responses to Questions

**Comments to the Author**

1. Is the manuscript technically sound, and do the data support the conclusions?

Reviewer #1: Yes

Reviewer #2: Partly

2. Has the statistical analysis been performed appropriately and rigorously? 

Reviewer #1: Yes

Reviewer #2: No

3. Have the authors made all data underlying the findings in their manuscript fully available?

Reviewer #1: Yes

Reviewer #2: Yes

4. Is the manuscript presented in an intelligible fashion and written in standard English?

Reviewer #1: Yes

Reviewer #2: Yes

5. Review Comments to the Author

Reviewer #1: The paper has been reviewed before, and the authors has addressed most issues of the manuscript. Below are a few comments for the authors to consider:

1. Please clarify the reference level for HR of non-binary variables in the abstract.

2. The font of number in table 5 is inconsistent with the font of number in other tables.

3. In the data source section, the authors mentioned that “all cases of adult migraineur patients, irrespective of gender and ethnicity, who visited the hospitals were selected for this study on the following specific criteria including … (c) all patients eligible for analysis were followed up for minimum 12 months”. The time in the study should not be one of the eligibility criteria for the survival analysis as patients who dropped out could still provide information about the event of interest by right censoring.

4. In the statistical analysis section, please clarify the meaning of the sentence “Therefore we used the data of those patients who completed our minimum follow-up study period (1 year follow-up) or the event data for survival analysis …”.

Reviewer #2: The manuscript has been much improved, especially the statistical analysis section. In the method section, the authors stated that they used the data of those patients who completed our minimum follow-up study period (1 year follow-up) or the event date for survival analysis while the data of patients who drop-out during study due to any reason or didn’t experienced event during whole study period were censored in survival analysis. This description is a bit confusing to me. If the minimum follow-up study period is an inclusion criterion, patients with less than 1-year follow-up would be excluded. Please consider clarifying the patients included in the analysis.

Also, some terms used in the manuscript are still confusing to me. For example, the title of table 6 is "Statistical analysis of migraineurs adherence towards treatment using chi-square and multiple linear cox regression analysis". Although a proportional Cox regression model can be converted to a linear regression model, the term of multiple linear cox regression analysis is confusing because I believe a proportional hazards model is used in the study and the model was not converted to a linear model.

In addition, I found a few grammar mistakes. For example, in Line 21 on page 17, "one of the study concluded" should be "one study concluded" or "one of the studies concluded". Similar grammar mistakes can be found elsewhere in the manuscript. Please also consider changing all p-values that are 0.000 to p<0.001. Theoretically, p-values cannot be equal to zero.

6. PLOS authors have the option to publish the peer review history of their article (what does this mean?). If published, this will include your full peer review and any attached files.

Reviewer #1: No

Reviewer #2: No

---

## [Author Response · Author response to Decision Letter 0]

22 Jan 2021

THE EDITORIAL OFFICE Dated 15-01-2021

PLOS ONE

Corrections in the manuscript entitled “Analysis of treatment cost and persistence among migraineurs: A two-year retrospective cohort study in Pakistan (Ms. Ref. No.: PONE-D-20-33485)”

Dear Editor,

First of all, authors are thankful to you for your useful comments and suggestions about our manuscript and giving us an opportunity to addressed reviewer’s comments with positive detailed corrections. The authors have modified the manuscript accordingly and the detailed corrections are listed below point by point. 

Response to Reviewer comments:

Reviewer # 1: 

1. Please clarify the reference level for HR of non-binary variables in the abstract.

Response to Reviewer comment no. 1: The reference level for hazard ratios of non-binary variables has been clarified in the abstract.

Abstract

Pairwise comparisons from Cox regression/hazards ratio were used to assess the predictors of persistence with the reference category of non-binary variables i.e. hazard ratio = 1 for low frequency migraineurs and NSAIDs users. 

2. The font of number in table 5 is inconsistent with the font of number in other tables.

Response to Reviewer comment no. 2: The font size has been corrected.

3. In the data source section, the authors mentioned that “all cases of adult migraineur patients, irrespective of gender and ethnicity, who visited the hospitals were selected for this study on the following specific criteria including … (c) all patients eligible for analysis were followed up for minimum 12 months”. The time in the study should not be one of the eligibility criteria for the survival analysis as patients who dropped out could still provide information about the event of interest by right censoring. 

Response to Reviewer comment no. 3: We authors are fully agreed with the reviewer concern about this point. Basically this criterion was added by the recommendation of previous reviewer in his first round of review. However, in this study minimum 12 months follow-up criteria was only consider for the determination of patient persistence (273 days continuation) with migraine treatment while all patients data which were initially enrolled in our study (2043 N) including dropped out patients during our study period were also already included during survival analysis as indicated in Kaplan Meier curves and HR model. 

In addition, correction has been made for this specific criterion for further clarification.

Similarly, it has already been mentioned in statistical analysis that the dropped out patients were also included in survival analysis as right censored data. 

The sentence has also been added in abstract as well. 

Data source

All cases of adult migraineur patients, irrespective of gender and ethnicity, who visited the hospitals, were selected for this study on the following specific criteria including (a) patients who gave permission & were willing to join this study and (b) a confirmed diagnosis of migraine and at least one episode of migraine during last one month. However, patients who were completed the minimum followed up period of 12 months or had an experienced of any event (discontinuation or switches) during this period were included for the determination of treatment persistence i.e. 273 days continuation.

Statistical analysis

The Cox proportional hazard regression model is often used to analyze covariate information that changes over time, with the hazard proportional. Therefore we used the data of all study patients including the data of patients who drop-out during study due to any reason as right censored data in survival analysis.

Abstract

The minimum follow up period for each migraineur was about 12 months for persistence analysis while dropped-out patients data were also included in survival analysis as right censored data.

4. In the statistical analysis section, please clarify the meaning of the sentence “Therefore we used the data of those patients who completed our minimum follow-up study period (1 year follow-up) or the event data for survival analysis …”.

Response to Reviewer comment no. 4: This sentence was also added by the recommendation of previous reviewer in his first round of review. Now sentence has been modified and more clarified as per reviewer recommendation.

Statistical analysis

Therefore we used the data of all study patients including the data of patients who drop-out during study due to any reason as right censored data in survival analysis.

Reviewer # 2: 

1. In the method section, the authors stated that they used the data of those patients who completed our minimum follow-up study period (1 year follow-up) or the event date for survival analysis while the data of patients who drop-out during study due to any reason or didn’t experienced event during whole study period were censored in survival analysis. This description is a bit confusing to me. If the minimum follow-up study period is an inclusion criterion, patients with less than 1-year follow-up would be excluded. Please consider clarifying the patients included in the analysis.

Response to Reviewer comment no. 1: Similar concern is also raised by first reviewer. We authors are fully agreed with the reviewers concern about this point. Basically this criterion was added by the recommendation of previous reviewer in his first round of review. However, in this study minimum 12 months a follow-up criterion was consider for the determination of patient persistence (273 days continuation) with migraine treatment. Therefore, sample size had already been reduced to 1597 N from 2043 N in all initial tables. However, all patients’ data which were initially enrolled in our study (2043 N) including dropped out patients during our study period were also included during survival analysis as indicated in Kaplan Meier curves and HR model. 

In addition, correction has been made for this specific criterion for further clarification.

Similarly, it has already been mentioned in statistical analysis that the dropped out patients were also included in survival analysis as right censored data. 

The sentence has also been added in abstract as well. 

Data source

All cases of adult migraineur patients, irrespective of gender and ethnicity, who visited the hospitals, were selected for this study on the following specific criteria including (a) patients who gave permission & were willing to join this study and (b) a confirmed diagnosis of migraine and at least one episode of migraine during last one month. However, patients who were completed the followed up period of 12 months were included for the determination of treatment persistence i.e. 273 days continuation.

Statistical analysis

The Cox proportional hazard regression model is often used to analyze covariate information that changes over time, with the hazard proportional. Therefore we used the data of all study patients including the data of patients who drop-out during study due to any reason as right censored data in survival analysis.

Abstract

The minimum follow up period for each migraineur was about 12 months for persistence analysis while dropped-out patients data were also included in survival analysis as right censored data.

2. Also, some terms used in the manuscript are still confusing to me. For example, the title of table 6 is "Statistical analysis of migraineurs adherence towards treatment using chi-square and multiple linear cox regression analysis". Although a proportional Cox regression model can be converted to a linear regression model, the term of multiple linear cox regression analysis is confusing because I believe a proportional hazards model is used in the study and the model was not converted to a linear model.

Response to Reviewer comment no. 2: Authors really thankful to reviewer for clarifying this point. The term has been corrected as per reviewer recommendation.

Table 6: Statistical analysis of migraineurs persistence towards treatment using chi-square and cox proportional hazard regression analysis

3. In addition, I found a few grammar mistakes. For example, in Line 21 on page 17, "one of the study concluded" should be "one study concluded" or "one of the studies concluded". Similar grammar mistakes can be found elsewhere in the manuscript. 

Response to Reviewer comment no. 3: All grammar mistakes have been corrected now. 

However, persistence with prevalent patients showed different results in comparison with incident patients and approximate 70% of prevalent patients exhibit non-persistence behavior with their treatment which could be due to avoid some potential hazards associated with drug and one study concluded the same results of poor persistence with prevalent / prophylactic treatment [9]. 

4. Please also consider changing all p-values that are 0.000 to p<0.001. Theoretically, p-values cannot be equal to zero.

Response to Reviewer comment no. 4: All p-values have been corrected as per reviewer recommendation.

---

## [Decision Letter · Decision Letter 1]

3 Mar 2021

PONE-D-20-33485R1

Analysis of treatment cost and persistence among migraineurs: A two-year retrospective cohort study in Pakistan

PLOS ONE

Dear Dr. Asghar,

Thank you for submitting your manuscript to PLOS ONE. After careful consideration, we feel that it has merit but does not fully meet PLOS ONE’s publication criteria as it currently stands. Therefore, we invite you to submit a revised version of the manuscript that addresses the points raised during the review process. These are only minor changes, so we are confident that you can make them soon.

We look forward to receiving your revised manuscript.

Kind regards,

Claudia Sommer

Academic Editor

PLOS ONE

Journal Requirements:

Reviewers' comments:

Reviewer's Responses to Questions

**Comments to the Author**

1. If the authors have adequately addressed your comments raised in a previous round of review and you feel that this manuscript is now acceptable for publication, you may indicate that here to bypass the “Comments to the Author” section, enter your conflict of interest statement in the “Confidential to Editor” section, and submit your "Accept" recommendation.

Reviewer #1: All comments have been addressed

Reviewer #2: All comments have been addressed

2. Is the manuscript technically sound, and do the data support the conclusions?

Reviewer #1: Yes

Reviewer #2: Yes

3. Has the statistical analysis been performed appropriately and rigorously? 

Reviewer #1: Yes

Reviewer #2: Yes

4. Have the authors made all data underlying the findings in their manuscript fully available?

Reviewer #1: Yes

Reviewer #2: Yes

5. Is the manuscript presented in an intelligible fashion and written in standard English?

Reviewer #1: Yes

Reviewer #2: Yes

6. Review Comments to the Author

Reviewer #1: The authors have done an excellent and comprehensive job of amending this manuscript. I only have

two further comments relating to my previous review:

-Abstract (do not indent; must include OBJECTIVES, METHODS, RESULTS, CONCLUSION )

-In abstract, please rewrite sentence "..1597 “N” migraineurs...".

Reviewer #2: (No Response)

7. PLOS authors have the option to publish the peer review history of their article (what does this mean?). If published, this will include your full peer review and any attached files.

Reviewer #1: No

Reviewer #2: No

---

## [Author Response · Author response to Decision Letter 1]

3 Mar 2021

THE EDITORIAL OFFICE Dated 03-03-2021

PLOS ONE

Corrections in the manuscript entitled “Analysis of treatment cost and persistence among migraineurs: A two-year retrospective cohort study in Pakistan (Ms. Ref. No.: PONE-D-20-33485R1)”

Dear Editor,

First of all, authors are thankful to you for your useful comments and suggestions about our manuscript and giving us an opportunity to addressed reviewer’s comments. The authors have modified the manuscript accordingly and the detailed corrections are listed below point by point. 

Response to Reviewer comments:

Reviewer # 1: 

1. Abstract (do not indent; must include OBJECTIVES, METHODS, RESULTS, CONCLUSION) 

Response to Reviewer comment no. 1: Abstract has been structured as per reviewer recommendation.

Abstract

OBJECTIVES: The persistence pattern of anti-migraine drugs’ use among migraineurs is very low in the United States and different European countries. However, the cost and persistence of antimigraine drugs in Asian countries have not been well-studied. Hence, the present study aimed to evaluate the treatment cost and persistence among migraineurs in Pakistan. METHODS: Data from prescriptions collected from migraineurs who visited the Outpatient Department (OPD) of different public and private sector tertiary-care hospitals of Karachi, Pakistan were used to conduct this retrospective cohort study from 2017 to 2019. The minimum follow up period for each migraineur was about 12 months for persistence analysis while dropped-out patients data were also included in survival analysis as right censored data. Pairwise comparisons from Cox regression/hazards ratio were used to assess the predictors of persistence with the reference category of non-binary variables i.e. hazard ratio = 1 for low frequency migraineurs and NSAIDs users. Persistence with anti-migraine drugs was estimated using the Kaplan-Meier curve along with the Log Rank test. RESULTS: A total of 1597 patients were included in this study, 729 (45.6%) were male and 868 (54.3%) were female. Non-steroidal anti-inflammatory drugs (NSAIDs) were the most prescribed class of drug initially for all classes of migraineurs (26.1 %). Of them, 57.3% of migraineurs discontinued their treatment, 28.5% continued while 14.8% were switched to other treatment approaches. Persistence with initial treatment was more profound in female (58.8%) patients compared to males while the median age of continuers was 31 years. The total cost of migraine treatment in the entire study cohort was 297532.5 Pakistani Rupees ($1901.1). By estimating the hazard ratios (HR) using the cox regression analysis, it can be observed that patients with high frequency (HR, 1.628; 95%CI, 1.221-2.179; p<0.0001) migraine, depression (HR, 1.268; 95%CI, 1.084-1.458; p<0.0001), increasing age (HR, 1.293; 95%CI, 1.092-1.458; p<0.0001), combination analgesics (HR, 1.817; 95%CI, 0.841-2.725; p=0.0004) and prophylaxis drugs (HR, 1.314; 95%CI, 0.958-1.424; p<0.0001) users were at a higher risk of treatment discontinuation. However, patients with chronic migraine (HR, 0.881; 95%CI, 0.762-0.912; p=0.0002), epileptic seizure (HR, 0.922; 95%CI, 0.654-1.206; p=0.0002), other comorbidities (HR, 0.671; 95%CI, 0.352-1.011; p=0.0003) and users of triptan(s) (HR, 0.701; 95%CI, 0.182-1.414; p=0.0005) and triptan(s) with NSAIDs (HR, 0.758; 95%CI, 0.501-1.289; p<0.0001) had more chances to continue their initial therapy. CONCLUSION: Similar to western countries, the majority of migraineurs exhibited poor persistence to migraine treatments. Various factors of improved persistence were identified in this study.

2. -In abstract, please rewrite sentence "..1597 “N” migraineurs...".

Response to Reviewer comment no. 2: The sentence has been corrected.

A total of 1597 patients were included in this study, 729 (45.6%) were male and 868 (54.3%) were female.

---

## [Editor Report · Decision Letter 2]

5 Mar 2021

Analysis of treatment cost and persistence among migraineurs: A two-year retrospective cohort study in Pakistan

PONE-D-20-33485R2

Dear Dr. Asghar,

We’re pleased to inform you that your manuscript has been judged scientifically suitable for publication and will be formally accepted for publication once it meets all outstanding technical requirements.

Kind regards,

Claudia Sommer

Academic Editor

PLOS ONE

---

## [Editor Report · Acceptance letter]

16 Mar 2021

PONE-D-20-33485R2 

Analysis of treatment cost and persistence among migraineurs: A two-year retrospective cohort study in Pakistan 

Dear Dr. Asghar:

I'm pleased to inform you that your manuscript has been deemed suitable for publication in PLOS ONE. Congratulations! Your manuscript is now with our production department. 

Kind regards, 

on behalf of

Prof. Dr. Claudia Sommer 

Academic Editor

PLOS ONE